# Manganese catalyzed oximation of hydrocarbons to oximes

Menghui Song[1,4], Hong Li[1,4], Li Xie[2], Xiaoxin Zhang[2], Xiaotian Weilian[1], Rucao Wang[1], Alexander Steiner [3], Huaming Sun[1], Chao Wang [1], Jianliang Xiao [3] ✉ & Chaoqun Li [1] ✉

Oximes are used in many scientific and industrial domains, ranging from organic synthesis through biotechnology to materials science. Traditionally, their synthesis necessitates the use of hydrocarbons with specific functional groups, such as carbonyls, which are often more challenging and expensive to obtain compared to their non-functionalized counterparts. Here we introduce an approach that enables the synthesis of oximes from hydrocarbons via the oxidative oximation of methylene C-H bonds — the most prevalent molecular unit in the world of molecules. Under the catalysis of a manganese complex with hydrogen peroxide as the oxidant and hydroxylamine sulfate as the amine source, we demonstrate that a diverse array of molecules — from simple chemicals like propane and cyclohexane to complex compounds such as the antimalarial drug artemisinin — can be oximated at methylene C-H bonds with synthetically significant yields under mild conditions. The catalyst displays a good level of functionality tolerance and often predictable site selectivity in complex molecule settings. Our approach opens an avenue for oxime synthesis and is anticipated to have broad applications in the production of fine and commodity chemicals, bioactive molecules, and new materials.

Oximes play important roles in diverse fields of science and technology. For example, they are used as building blocks in organic synthesis, active components of drugs, food sweeteners, herbicides, linkers for the conjugation of biomolecules and biomaterials, and "chameleons" in plant metabolism (Fig. 1a; also see Figs. S-1 and S-2)[1–4]. The largest industrial application of oximes is in the manufacturing of nylon-6, nylon-12, and anti-skinning agents in paints[5]. In 2024, the global production capacity for nylon-6 is expected to exceed 8.8 million metric tons, while that for nylon-12 has reached hundred thousand tons[6].

Oximes can be synthesized by a variety of methods. These methods usually rely on the conversion of a pre-installed functionality, such as carbonyl and amino moieties (Fig. 1b)[1,2,7–10]. The vast majority of oximes are synthesized by oximation of carbonyl compounds with hydroxylamine, and this is particularly the case in large scale industrial

processes[1,2,5,11]. Because carbonyls, such as ketones, can be synthesized from the more easily available, more economic, saturated hydrocarbons, oximation of the latter would make the synthesis of oximes more economic and greener. The major industrial process of the synthesis of nylon-6 is illustrative (Fig. 1b)[5]. The process starts with aerobic oxidation of cyclohexane, yielding a mixture of cyclohexanone and cyclohexanol, referred to as KA oil, with a selectivity of 80–85%. To avoid overoxidation, the conversion is kept low, at 10–12%; thus large recycles of unreacted cyclohexane are necessary. The cyclohexanone is separated and the remaining cyclohexanol is dehydrogenated to cyclohexanone. Reaction of the cyclohexanone with hydroxylamine sulfate affords cyclohexanone oxime, which is then subjected to Beckmann rearrangement to form the nylon-6 precursor, caprolactam. Nylon-12 has been synthesized similarly. Clearly, this multistep process

[1]Key Laboratory of Applied Surface and Colloid Chemistry, Ministry of Education and School of Chemistry and Chemical Engineering, Shaanxi Normal University, Xi'an, China. [2]State Key Laboratory of Petroleum Molecular & Process Engineering, Sinopec Research Institute of Petroleum Processing, Beijing, China. [3]Department of Chemistry, University of Liverpool, Liverpool, UK. [4]These authors contributed equally: Menghui Song, Hong Li. ✉e-mail: jxiao@liverpool.ac.uk; lichaoqun@snnu.edu.cn

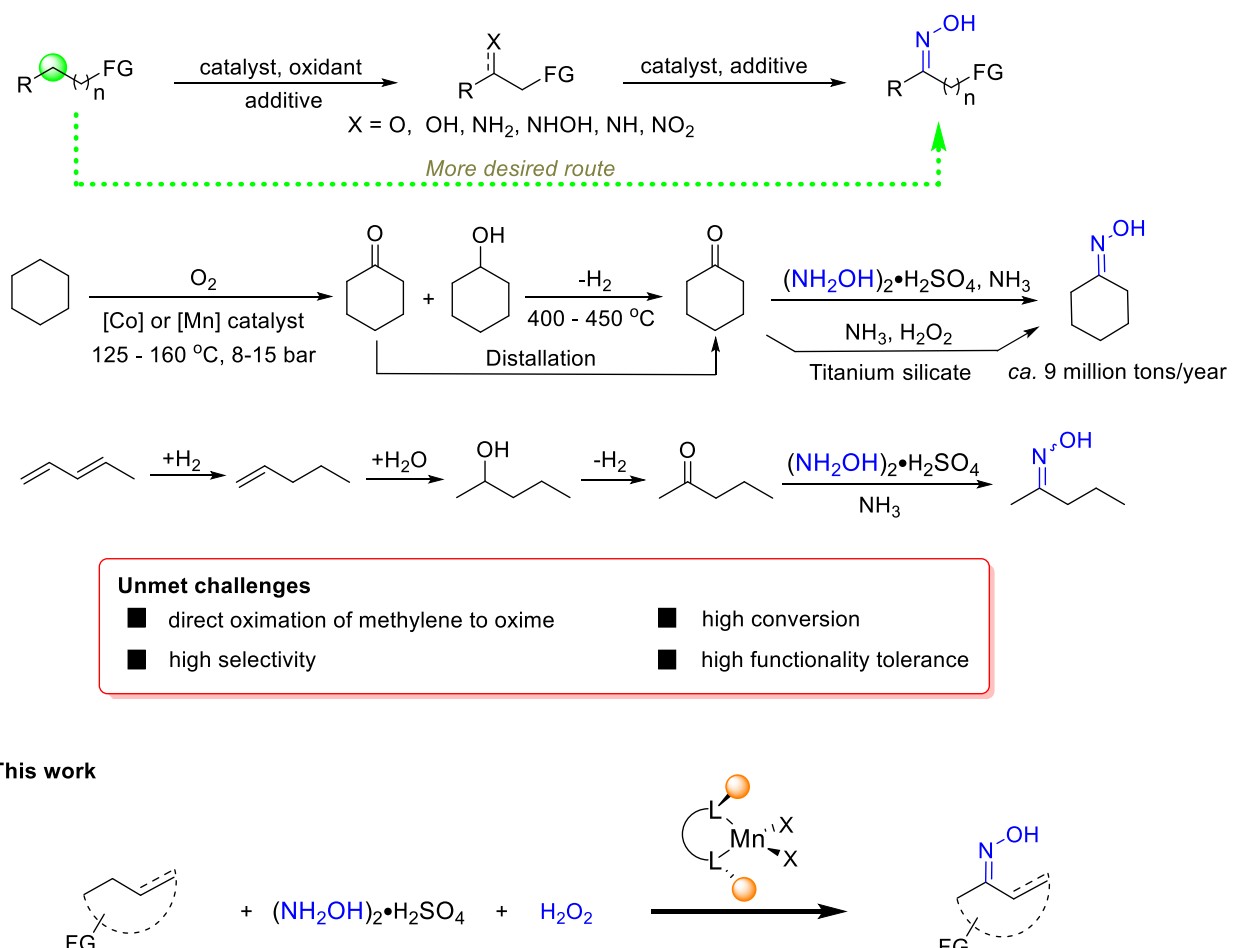

**Fig. 1 | Examples of industrial oxime products and their synthesis with established and new methods. a** Examples of oximes found in fine and commodity chemicals and drug molecules. **b** Common methods for oxime synthesis (FG can be a functional or nonfunctional group) and industrial routes for cyclohexanone and 2-pentanone oximes. The green line highlights a desired approach. **c** The route reported in this paper.

is inefficient and highly energy-intensive. In recent years, great progress has been made in improving the process—globally *ca.* 50% of the cyclohexanone oxime is now being synthesized by a new process involving ammoximation of cyclohexanone with $NH_3$ and $H_2O_2$[5,12]. More recently, synthesis of cyclohexanone oxime has been made possible via ammoximation with $H_2O_2$ in-situ generated from $H_2$ and $O_2$[13]. However, in all these processes, cyclohexanone, instead of

cyclohexane, is the hydrocarbon feedstock. An example of fine chemicals is seen in the synthesis of 2-pentanone oxime, an anti-skinning agent used in the coating industry (Fig. 1b). The process comprises again a series of reactions, starting with expensive 1,3-pentadiene (Fig. S-5).

Generating oximes from primary petrochemicals instead of their derivatives, oximation of the methylene C-H bonds in hydrocarbons

via oxidation presents an ideal approach for the construction of oximes (Fig. 1b)[14–18]. Furthermore, C-H bonds are ubiquitous in organic molecules. Selective oximation of such bonds would open a short-cut pathway for oxime synthesis, allowing oximes of diverse properties to be more easily accessible. This is challenging, however, because of (1) the strong C(sp$^3$)−H bond that renders the reaction kinetically and thermodynamically difficult (e.g. BDE of H-CH(CH$_3$)$_2$: 99 kcal/mol), (2) the presence of multiple methylene C-H bonds of similar BDEs that can lead to low site selectivity, (3) possible catalyst poisoning by the amine substrate and oxime product, and (4) functionalities in complex substrates that may not tolerate the oxidation conditions employed. Not surprisingly, there have been only a few reported reaction systems capable of converting inactive C−H bonds into oximes, but these either require harsh conditions or face poor selectivity issues. For instance, cyclohexanone oxime has been synthesized industrially by photonitrosation of cyclohexane with a mixture of NOCl and HCl under UV irradiation and by nitration of cyclohexane with HNO$_3$[5,19]. Both processes rely on highly corrosive nitrosating reagents and are energy inefficient. Radical nitrosation of cycloalkanes with 'BuONO via UV irradiation has also been reported, which gives a mixture of nitrosocycloalkanes and cycloalkanone oxime[20]. For fine chemical and pharmaceutical synthesis, such C-H oximation methods would be of only limited use, as they would not be expected to offer good chemo and site-selectivity and tolerate diverse functional groups. Thus, there remains a large space to improve the efficacy of direct C-H oximation of hydrocarbons for oxime synthesis.

Herein, we report a manganese catalyst-centered protocol that enables the oxidative oximation of a broad range of hydrocarbons with H$_2$O$_2$ and (NH$_2$OH)$_2$•H$_2$SO$_4$ under mild conditions, affording various acyclic and cyclic ketone oximes including the lactam precursors for nylons 5, 6 and 12 (Fig. 1c). Adding to the advantage of a one-pot reaction, manganese is one of the most abundant base metals, of low toxicity and biocompatible[21,22]. Manganese as well as iron-based catalysts have been intensively studied in selective oxidation reactions in recent years; however, none is known to catalyze the oximation of hydrocarbons[16,21–23].

## Results
### Reaction development
We have shown previously that manganese complexes bearing tetradentate amino ligands catalyze efficient benzylic oxidation of alkyl arenes to aryl ketones with H$_2$O$_2$, and if the alkyl chain bears a primary amino moiety, cyclic imines are generated via in-situ attack of the amine at the ketone[24]. Replacing the intramolecular amine moiety with an exogeneous hydroxylamine, we might be able to obtain an oxime under the manganese catalysis. However, both the substrate NH$_2$OH and product oximes are known to coordinate to metals[2] and could thus easily deactivate a metal catalyst. In addition, oximes are difficult to be protonated at the nitrogen atom (p$K$a < 1)[25]; thus, if coordination of an oxime is to be suppressed, a strong acid would be necessary. With these considerations in mind, we chose at the outset the common industrial feedstock hydroxylammonium sulfate as the oxime source (Fig. 2). We started by examining the oxidative oximation of cyclododecane 1 with H$_2$O$_2$ and (NH$_2$OH)$_2$•H$_2$SO$_4$, the oxime product of which is the precursor to nylon-12. On screening a wide range of reaction variables, we found that the manganese complex rac-**C1**, i.e. **C1** containing a racemic pyridine-bipiperidine ligand (PYBP), catalyzes the efficient oxidative oximation of cyclododecane under the conditions shown (Fig. 2a). The yield of the oxime product **1a** was found to be influenced by a variety of parameters, including catalyst, solvent, amount of acidic additive and the ratio of H$_2$O$_2$/substrate (Table S-2). Thus, the PYBP ligands with electron-donating, bulky substituents tend to give a higher oxime yield, with the one bearing a 5,5'-triisopropylsilyl group being most effective (rac-**C1**) (Fig. 2a). Of note is that the meso-**C2** and meso-**C3** complexes showed little activity while meso-**C1**

compares well with rac-**C1**. Acidic conditions and the use of the protonated hydroxylamine are essential; no reaction took place under neutral conditions or when free hydroxylamine was used (Table S-2, and SI, Section 6.7). Under the conditions established, the cyclododecanone oxime **1a** was obtained in an isolated yield of 91%. Compared with the related ketonization reactions, the yield of **1a** is higher (Fig. S-6; also see SI, Section 6.6).

The formation of the oxime **1a** was found to proceed via the intermediacy of cyclododecanone **1-one** resulting from the oxidation of **1** with H$_2$O$_2$. Monitoring the reaction with NMR showed that **1-one** was formed progressively. Subsequent basification of the reaction mixture then afforded **1a**, which was necessary as hydroxylamine is protonated (p$K$a 5.9) and thus might not be able to attack the ketone[26]. Indeed, **1-one** was isolated in 85% yield when the basification was omitted (SI, Section **6.6**). However, further study into the oxidation of cyclododecane, cyclohexane, and derivatives with GC and NMR indicates that oxime was formed from the ketone almost immediately after its formation from the hydrocarbon without basification, and the resulting oxime could be in equilibrium with its solid form, sulfate salt, which releases the free oxime upon basification (SI, Sections 6.2–6.3). The oximation of **1-one** appears to be much faster than the formation of **1-one** (SI, Section 6.5; Table S-16). Figure 2b shows the time course of the formation of **1a** from **1**.

Little multiple oxidations were noted in the reaction. This is presumably due to the ketone moiety introduced imparting a deactivating effect on the remaining methylene C-H bonds toward oxidation by electrophilic Mn(V) = O species[27]. However, the oximation also plays an important role, as a mixture of polyketone products was generated in the absence of hydroxylammonium, with **1-one** isolated in only 38% yield (Fig. 2c). Like carbonyl, the oxime group is deactivating, as shown by the oxidation of **1a** vs **1-one**, the former being markedly resistant toward further oxidation (SI, Section 6.4). In addition, comparing the disappearance rate of **1** (c.f. Fig. 2b, c) suggests that the hydroxylammonium ion may suppress the oxidation ability of the catalyst, rendering overoxidation less likely. Further studies showed that the hydroxylammonium ion also exerts a significant effect on the chemo and site-selectivity in the oxidation of more complex molecules (for examples, see SI, Sections **6.9**–**6.10**). While the detailed mechanism of the hydroxylammonium effect remains to be elucidated, our preliminary investigations suggest that the introduction of the hydroxylammonium cation affects the formation of high valent manganese-oxygen species from rac-**C1** and it could be involved in the step of C-H bond oxidation (SI, Sections 6.11–6.13; for a suggested mechanism, see Section 6.14).

### Oximation of unactivated C(sp$^3$)-H bonds
Under the optimized conditions, a range of nonfunctionalized cyclic and acyclic alkanes have been converted to the corresponding ketone oximes (1.0 mmol scale, Fig. 3a). Cyclic oximes ranging from cyclopentanone to cyclododecanone oximes were obtained in high yields with little overoxidation observed. However, the larger sized cyclopentadecanone oxime was obtained in a lower 74% yield, where a small amount of multi-oxidation products was observed. The bicyclic norbornane **7** was oximated at a methylene site with no oxidation at the tertiary C-H position observed. For the more challenging acyclic alkanes, a larger amount of HOAc was required to achieve a satisfied yield (Tables S-4 and S-5). Under such conditions, acyclic alkanes ranging from propane to pentane were oximated exclusively at the methylene C-H bond next to the terminal methyl group, as would be expected from the steric accessibility and bond strength effect. For longer alkanes, e.g., **11a** and **12a**, regio-isomers (all mono-oxime products) were formed because of diminishing steric and electronic differences between the methylene units. However, oximation of the terminal methylene units remained dominating (see SI for more examples), and the regioselectivity of rac-**C1** in oxidizing **11** and **12** is

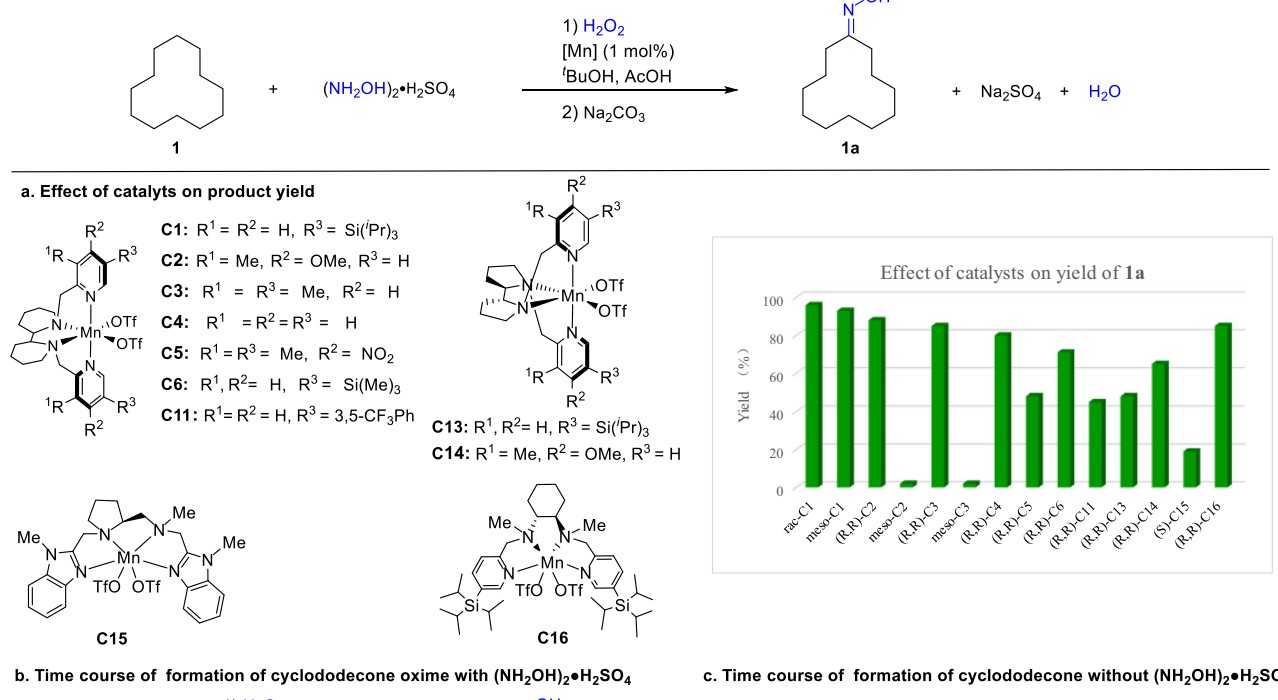

### a. Effect of catalysts on product yield

C1: $R^1 = R^2 = H$, $R^3 = Si(^iPr)_3$
C2: $R^1 = Me$, $R^2 = OMe$, $R^3 = H$
C3: $R^1 = R^3 = Me$, $R^2 = H$
C4: $R^1 = R^2 = R^3 = H$
C5: $R^1 = R^3 = Me$, $R^2 = NO_2$
C6: $R^1, R^2 = H$, $R^3 = Si(Me)_3$
C11: $R^1 = R^2 = H$, $R^3 = 3,5\text{-}CF_3Ph$

C13: $R^1, R^2 = H$, $R^3 = Si(^iPr)_3$
C14: $R^1 = Me$, $R^2 = OMe$, $R^3 = H$

### b. Time course of formation of cyclododecone oxime with $(NH_2OH)_2 \bullet H_2SO_4$

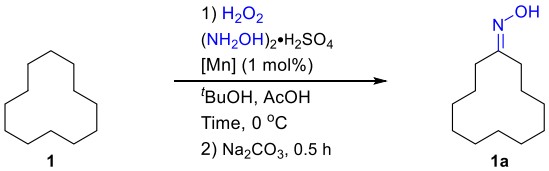

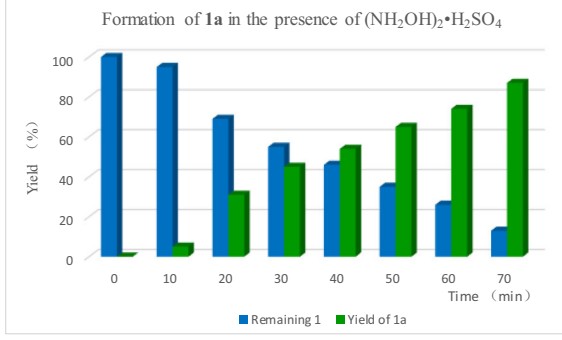

### c. Time course of formation of cyclododecone without $(NH_2OH)_2 \bullet H_2SO_4$

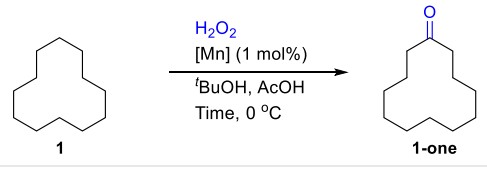

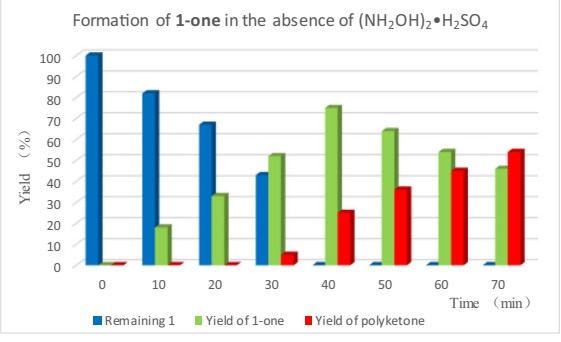

**Fig. 2 | Oximation of cyclododecane 1 via oxidation with $H_2O_2$ under manganese catalysis. a** Effect of catalysts on the oximation. Reaction conditions: **1** (0.5 mmol), catalyst (1 mol%), $(NH_2OH)_2 \bullet H_2SO_4$ (1.0 mmol) and AcOH (0.6 mL) dissolved in $^tBuOH$ (1.0 mL), and then $H_2O_2$ (2.5 mmol) in 1.0 mL of $^tBuOH$ introduced with a syringe pump over 1 h under stirring at 0 °C. After an additional 0.5 h, the solution was quenched with $Na_2SO_3$ and then basified with $Na_2CO_3$ for 0.5 h at 0 °C. **b** Time course of the formation of oxime **1a** with *rac*-**C1** under conditions given. **c** Time course of the formation of ketone **1-one** with *rac*-**C1** under conditions given. The yield of polyketone products is calculated from the yields of **1-one** and remaining **1**.

generally higher than that of other catalysts in related ketonization reactions (Fig. S6)[28–31]. Larger scale oximation and lower catalyst loading were also demonstrated, as seen with cyclododecane (Fig. 3a; also see Table S-4 and Section 10). The ability to oximate these unfunctionalized alkanes is remarkable. Traditional methods of accessing such oximes, when starting from cheap hydrocarbon substrates, are usually multi-step reactions under harsh conditions (Figs. S-3–S-5).

Functionalized cyclic alkanes were next examined (Fig. 3b). Good yields were generally obtained across the ring systems. For the five-membered ring substrates, oxidation took place highly selectively at the remote methylene site. Regio-isomers were seen in substituted six and seven-membered ring compounds (for their separation, see SI, Section **12**). For the seven-membered, the δ C-H bond appears to be more prone to undergo oxidation, as would be expected based on the decreasing

deactivating effect of the electron-withdrawing substituent[27,28,32]. For the six- membered, the γ position is slightly favored when a statistic correction is applied. This may be ascribed to the alleviation of the substituent-imposed 1,3-diaxial strain upon HAT at this site[28]. An exception is seen in the acetate substituted **16a**, where the γ site is less favored. This may be due to a higher axial population of the acetate group ($\Delta G_{eq/ax} = 0.8$ kcal/mol at 193 K)[33] than other substituents, which disfavors γ oxidation because of increased steric clash with the catalyst. In both cases, the yields of the phthalimide derivatives are significantly lower (**19a, 22a**), reflecting the strong deactivating role of the phthalimide moiety[27,34,35]. The norbornane **24** only afforded the δ oxime **24a** as a result of the steric and electronic effect of the substituent. The yields of and selectivities to these oxime products are comparable to those reported for the related ketones; they are, however, lower than those observed when the oxidation was carried out in perfluorinated

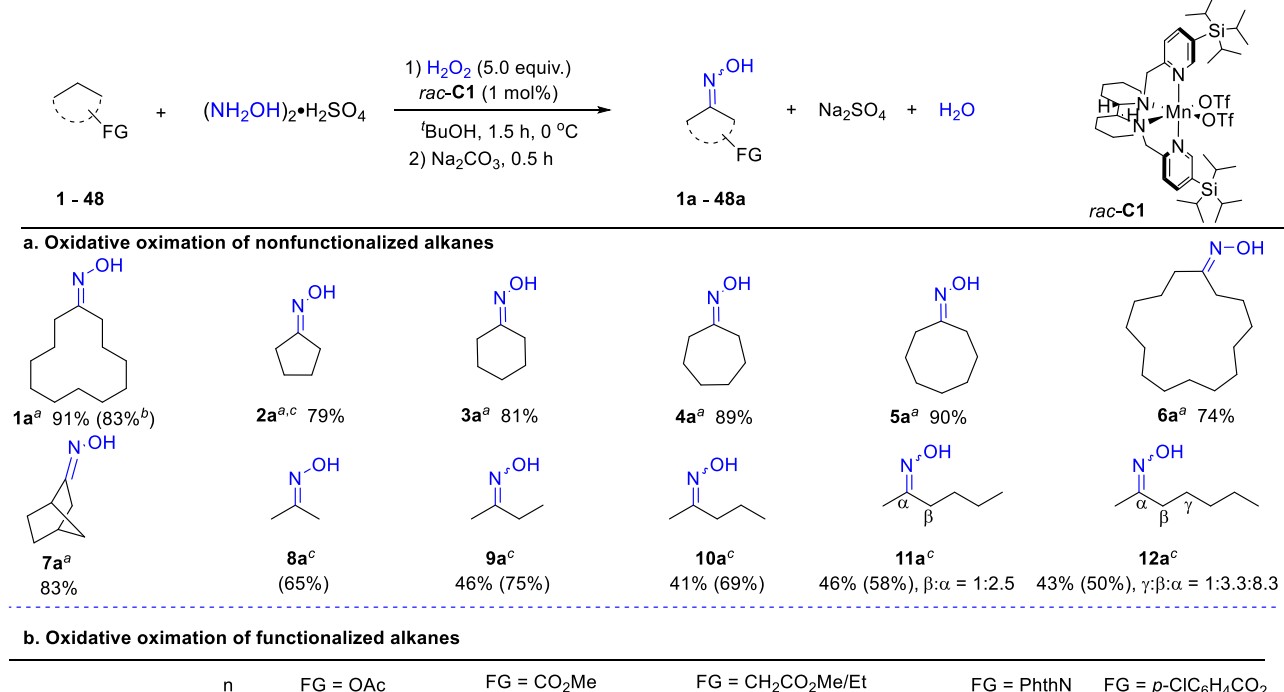

**a. Oxidative oximation of nonfunctionalized alkanes**

**b. Oxidative oximation of functionalized alkanes**

| | n | FG = OAc | FG = CO₂Me | FG = CH₂CO₂Me/Et | FG = PhthN | FG = p-ClC₆H₄CO₂ |
|---|---|---|---|---|---|---|
| | 0 | **13a**, 75% | **14a**, 81% | **15a** (Me), 83%, β:γ = 1:4 | / | / |
| | 1 | **16a**, 81% γ:δ = 1.5:1 | **17a**, 86% γ:δ = 2.7:1 | **18a** (Et), 87%, γ:δ = 2.5:1 | **19a**, 43% γ:δ = 2.5:1 | / |
| | 2 | **20a**, 80% γ:δ = 1:2 | **21a**, 84% γ:δ = 1:1.3 | / | **22a**, 60% γ:δ = 1:2.3 | **23a**, 75% γ:δ = 1:1.7 |

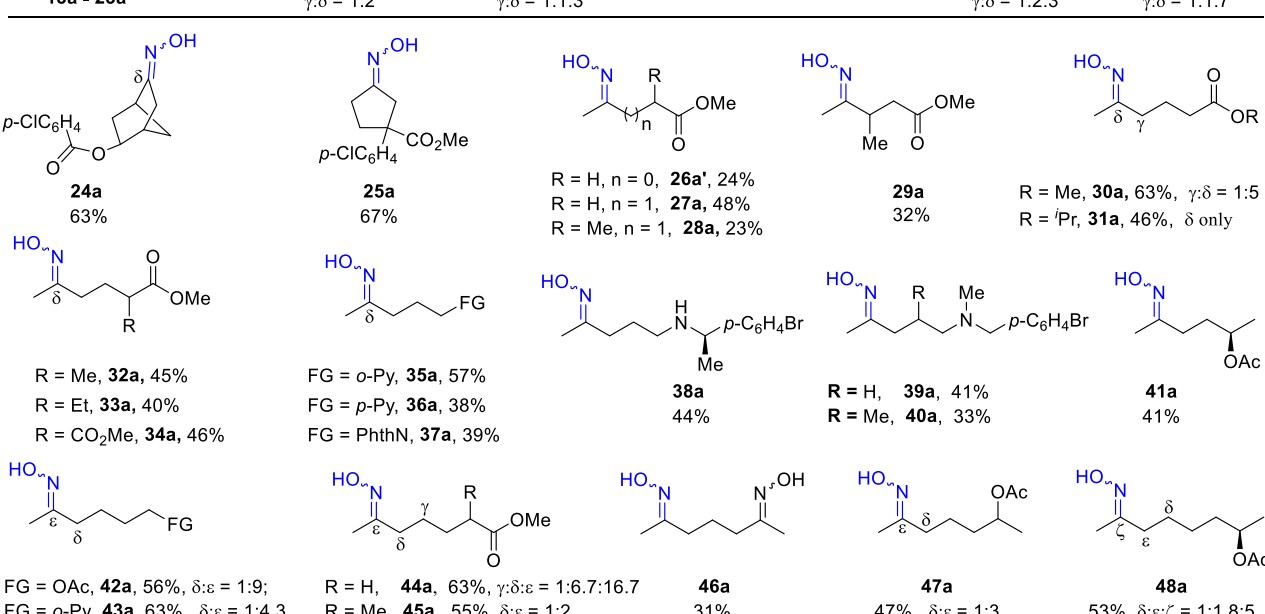

**Fig. 3 | Oximation of unactivated hydrocarbons with *rac*-C1.** General reaction conditions for **a**, **b**: substrate (1.0 mmol), *rac*-C1 (1 mol%), (NH₂OH)₂•H₂SO₄ (2.0 mmol) and AcOH (2.8 mL) dissolved in ᵗBuOH (2.0 mL), and then H₂O₂ (5.0 mmol) in 2 mL of ᵗBuOH introduced with a syringe pump over 1 h under stirring at 0 °C. After stirring for an additional 0.5 h, the solution was quenched with Na₂SO₃ and then basified with Na₂CO₃ for 0.5 h at 0 °C. Isolated yield reported (GC yield in parentheses). ᵃAcOH (1.2 mL). ᵇ10 mmol scale affording 1.64 g of **1a**, 83% isolated yield. ᶜYields based on hydroxyamine sulfate; hydrocarbon in excess.

alcohols[27]. Note that all the oximes exist in *E* and *Z* forms except for those having plane symmetry (ignoring the hydroxy unit).

Acyclic alkanes bearing various functionalities were also oximated (Fig. 3b). Particularly notable is the oximation of pyridine, amine, and oxime-bearing alkanes (**35a, 36a, 38a-40a, 43a, 46a**), the tolerance of which is likely due to the protonation of the heteroatoms that prevents their coordination to and hence the deactivation of the metal center. The protonation-elicited polarity reversal effect[27] may have also contributed to the higher site selectivity. Surprisingly somehow, with only one methylene unit separating the reacting C-H bond and the deactivating ester, methyl butyrate also underwent the reaction, affording a cyclized isoxazoline **26a'**, albeit in a low yield. A comparison of the

yields of oximes reveals the directing effect on site selectivity of electron-withdrawing substituents, which deactivate the proximal C-H bonds toward oxidation[27,28,32]. Thus, when the most remote methylene unit is less than 3-4 carbons away from the substituent, oxidation occurred selectively at that site. However, when the methylene is further away, oxidation of the proximal methylene units took place, resulting in regio-isomeric mono-oxidation products (e.g., **27a** vs **30a**, **41a** vs **47a**. Also see Table S-12). Similar effects have been noted before[27–29]. A theoretical study of *n*-dodecane bearing a terminal electron-withdrawing group (-NH$_3^+$) shows that the positive charge on the methylene hydrogen atoms stops varying after the fifth carbon[36]. Comparing the yields of and selectivities to **30a** and **46a** indicates that the oxime is more deactivating and hence more directing than a methyl ester (also see SI, Section **6.4**). The site selectivity and product yield are also affected sterically. Thus, making the substrate bulkier, regardless in the chain or by the functionality, increased the selectivity to the remote methylene, albeit at the expense of the oxime yield (*c.f.* **27a** vs **28a**, **30a** vs **31a** and **32a**). This likely stems from possible steric repulsion between the ligand and the approaching substrate during oxidation[28]. Although the yields of some of these functionalized oximes tend to be low, the reaction may still offer advantages in the context of complex molecule or nature product synthesis.

## Oximation of activated C($sp^3$)-H bonds

Prompted by the utility of allylic oxidation in natural product synthesis[37,38], we next attempted the oximation of allylic substrates. For this transformation involving activated methylene C-H bonds, the sterically less demanding *rac*-**C2** was shown to be more productive, probably partly due to the substates being more rigid and/or side reactions (Tables S-7 and S-9). As shown in Fig. **4**a, a range of acrylate derivatives were oximated. Although the ester unit is deactivating, the low BDE of the allylic C-H bond makes it the primary oximation site (e.g., BDE of the allylic C-H bond in cyclohex-1-ene-carboxylic acid: 88.4 kcal/mol)[39]. Thus, no site selectivity issue was noted in forming **49a**–**51a**. When there are methylene units three carbons remote from the allylic position, oxidation of these units was observed (**52a**–**54a**). However, allylic oxidation remained dominating. Epoxidation was noted as a minor reaction in some cases (SI, Table S-9), and surprisingly, ongoing from **64** to **64a**, the olefin unit was also epoxidized, albeit in a low yield probably due to the product being less stable (<5% of **64** recovered). As with alkane oximation (Fig. **3**), sterically demanding and strong electron-withdrawing substituents lower the product yield (e.g. **49a** vs **55a**, **50a** vs **56a**, and **59a** vs **61a**). Partial oxidation of a tertiary C-H bond to a hydroxy moiety was noted in addition to allylic oximation (**62a-4'-OH**). Acrylamides are also feasible (**66**–**68**). However, the oxidation of the primary amide **66** afforded an epoxide **66b** as the main product (**66b**, 34% yield; SI, Section **12**), and in the case of **67**, oxidative dealkylation to give **66a** was also observed. Notably, although highly strained, the cyclopropyl moiety in **68** survived. We also examined a propargylic substate **69**, which was oximated in a low yield of 21%, mainly due to product instability.

Oxmiation of benzylic substrates was also demonstrated (Fig. **4**b). *rac*-**C2** was again shown to be more productive than *rac*-**C1** (Table S-10). Benzylic oximes were obtained in yields generally higher than those of acyclic alkanes and acrylate derivatives, reflecting the strong activating effect of the aryl group. Notably, electron-withdrawing *para*-substituents, such as nitro and chloride, on the aryl rings pose little effect on the oxime yield, while the electron-donating methoxy did not cause significant competitive aromatic oxidation[40,41]. Also notable is the exclusive benzylic site selectivity, regardless of the presence of other methylene units (e.g. **81a**, **82a**, **94a**). Whilst the pyridine oximes **95a** and **96a** were obtained in low yields, possibly due to deactivation caused by the nitrogen being protonated, the pyridine-fused analogs **97a** and **98a** were isolated in good yields. The high yield may stem from a lower BDE of the benzylic C-H bond in an aryl-fused alkane. The

protonation-elicited deactivation is more pronounced when the nitrogen is at the *ortho* or *para* position due to the resonance effect (**95a**, **99a**). Encouragingly, a chiral 3-amino benzyl oxime **100a** was obtained in high yield, pointing to the potential of the protocol in enabling the synthesis of synthetically sought-after 1,3-difunctionalized chiral compounds. Note although benzylic oxidation reactions of functionalized alkyl arenes have been extensively studied, those featuring a pyridine group remain challenging[42,43].

## Synthetic applications

To probe the wider synthetic applicability of the protocol, we also carried out the oximation of a range of enantiomerically enriched substrates bearing multiple functionalities (Fig. **5**). As shown in Fig. **5**a, amino acid derivatives were tolerated, with the oxime yields varying with the position of the methylene unit to be oxidized. A good yield was obtained when the unit is three carbons away (*c.f.* **101a** and **102a**). The low yield of **101a** is likely a result of both electronic and steric effects. On the other hand, site isomers were formed when the alkyl chain becomes longer, e.g. **103a**, due to the decreased deactivating effect of the functionality. In contrast with **111**, the racemic amino acid derivative **112** was oximated to **112a** in a low yield. This may stem from the interplay of the C-H bond strength, steric hindrance of the substrate and the high ring strain of the product (*c.f.* strain energy of cyclobutanone and cyclopentanone: 28.7 vs 9.7 kcal/mol; BDE of C-H bond 97.8 in cyclobutane vs 95.6 kcal/mol in cyclopentane)[44]. When a substrate contains an activated methylene unit, site selectivity poses less a problem. Thus, the amino acid derivatives **115**–**119** were oximated at the allylic position, and **113** and **114** underwent highly efficient benzylic oximation. Note that **118a** was obtained in a higher yield when using (*R,R*)-**C2**, revealing the importance of chirality match between the substrate and catalyst, which in turn indicates possible steric interactions between the ligand and substrate during the oxidation. The enantiomeric purity of **115a** was measured and showed no erosion. Despite the low yields sometimes, these reactions allow multifunctionalized oximes to be accessed more directly and thus should be of value in the synthesis of nonnatural amino acid derivatives[3,45].

The high reactivity and selectivity displayed by the catalysts provide an opportunity to effect late-stage oximation of bioactive molecules (Fig. **5**b). As an example, sclareolide, a commercially available material in natural products synthesis, was oxidized to oxime **120b** in 56% isolated yield (total yield of isolated *E* and *Z* isomers) alongside small amounts of **120a** and **120c** under the catalysis of (*S,S*)-**C1**. The structures of (*E*)-**120a** and (*Z*)-**120b** have been determined by X-ray diffraction. Compared with the ketonization of **120** effected by other catalysts, the oxidation of **120** with (*S,S*)-**C1** shows a higher C2 regioselectivity. Another example is seen in the oxidation of the natural product methyl dehydroabietate **121**, which is isolated from spruce bark and has antibacterial activity. The oximation occurred exclusively at the benzylic methylene site, affording **121a** in single *E* form under the catalysis of (*R,R*)-**C2**. However, a small amount of the ketone intermediate **121b** remained. The oximation of the antimalaria drug artemisinin provides a further example. To date, late stage oxyfunctionalization of artemisinin has been limited to enzymatic[46,47] and iron catalysis[48,49]. Subjecting artemisinin **122** to the catalysis of (*R,R*)-**C1**, the oxime **122a** was isolated in 56% yield, with the oxidation occurring at the more electron rich and sterically more accessible methylene site. Note that using (*S,S*)-**C1** as catalyst resulted in the formation of an inseparable mixture, showing again the importance of chirality match. The oxime derivatives of **120**–**122** have been unknown so far. Installation of an oxime functionality to such molecules may alter their bioactivity and bring about unexpected new properties.

In summary, we have demonstrated that oximes can be installed via methylene C-H oxidation. The manganese complexes **C1** and **C2** catalyze the oxidative oximation of methylene C-H bonds in a wide variety of simple and complex molecules with benign H$_2$O$_2$ and

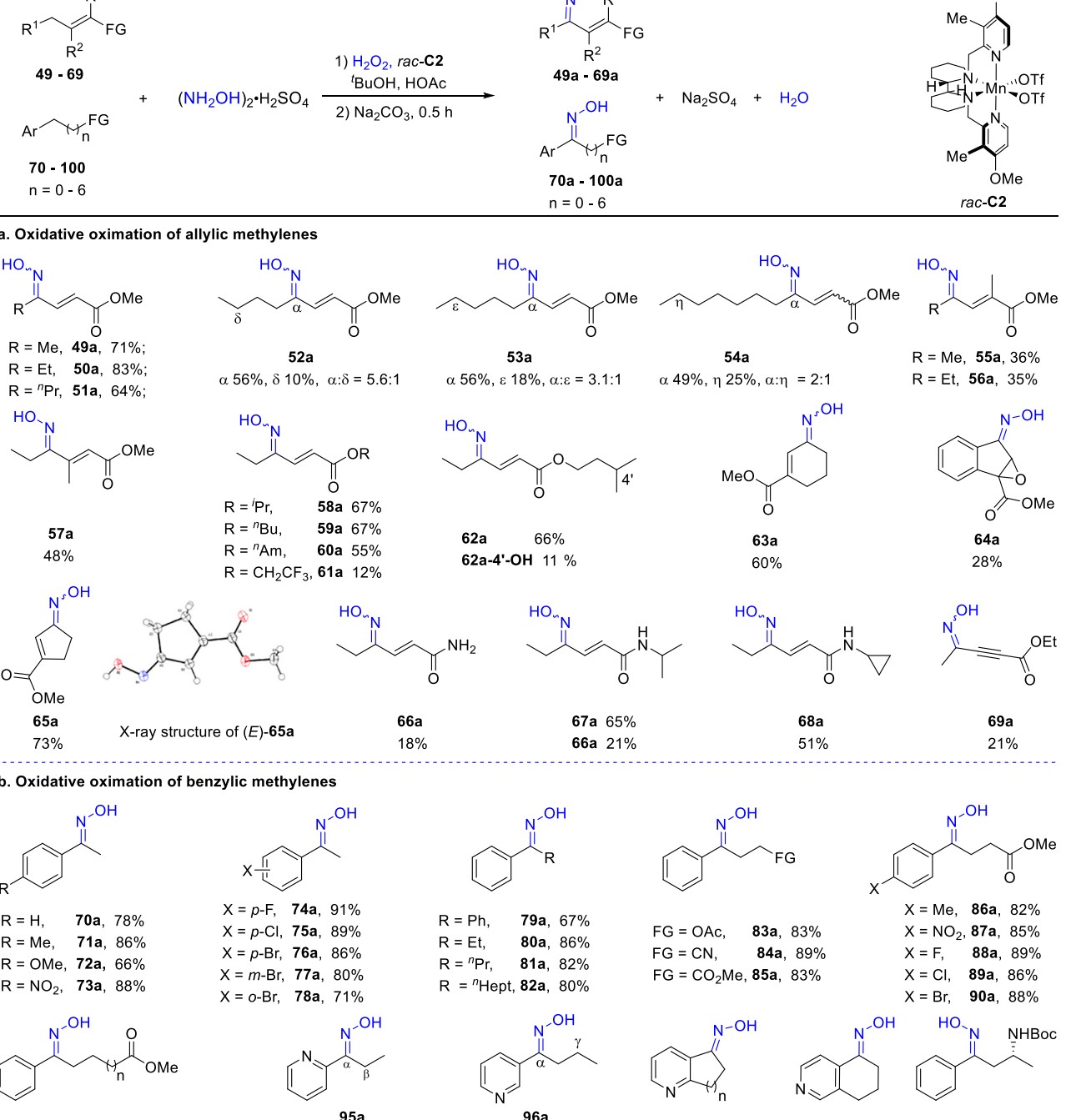

**Fig. 4 | Oximation of activated hydrocarbons with *rac*-C2. a** Conditions: substrate (1.0 mmol), *rac*-**C2** (1 mol%), (NH₂OH)₂•H₂SO₄ (2.0 mmol), and AcOH (2.0 mL) dissolved in ᵗBuOH (2.0 mL), and then H₂O₂ (5.0 mmol) in 2 mL of ᵗBuOH introduced with a syringe pump over 1 h under stirring at 0 °C. **b** Conditions: substrate (0.5 mmol), *rac*-**C2** (2 mol%), (NH₂OH)₂•H₂SO₄ (1.5 mmol), and AcOH (0.6 mL) dissolved in ᵗBuOH (1.0 mL), and then H₂O₂ (2.5 mmol) in 0.5 mL of ᵗBuOH introduced similarly to (**a**). In both cases, after stirring for an additional 0.5 h, the solution was quenched with Na₂SO₃ and then basified with Na₂CO₃ for 0.5 h at 0 °C (**a**) or 50 °C (**b**). Isolated yields reported.

economic (NH₂OH)₂•H₂SO₄, displaying a high level of, and often predictable, chemo and site selectivity and functional group tolerance. The electron-rich, bulky **C1** is more effective toward nonactivated C-H bonds, while the electron-rich but less bulky **C2** is better for activated variants. We anticipate the catalytic system to find applications in selective oximation of hydrocarbons and in the synthesis of fine and complex organic chemicals.

## Methods

### Representative procedure for oximation of unactivated hydrocarbons with *rac*-C1

*Rac*-**C1** (10.2 mg, 1.0 mol%), hydroxylamine sulfate (328.3 mg, 2.0 mmol, 2.0 equiv.), a hydrocarbon substrate (1.0 mmol, 1.0 equiv.), AcOH (2.8 mL for functionalized alkanes and nonfunctionalized acyclic alkane; 1.2 mL for nonfunctionalized cyclic alkanes) and ᵗBuOH

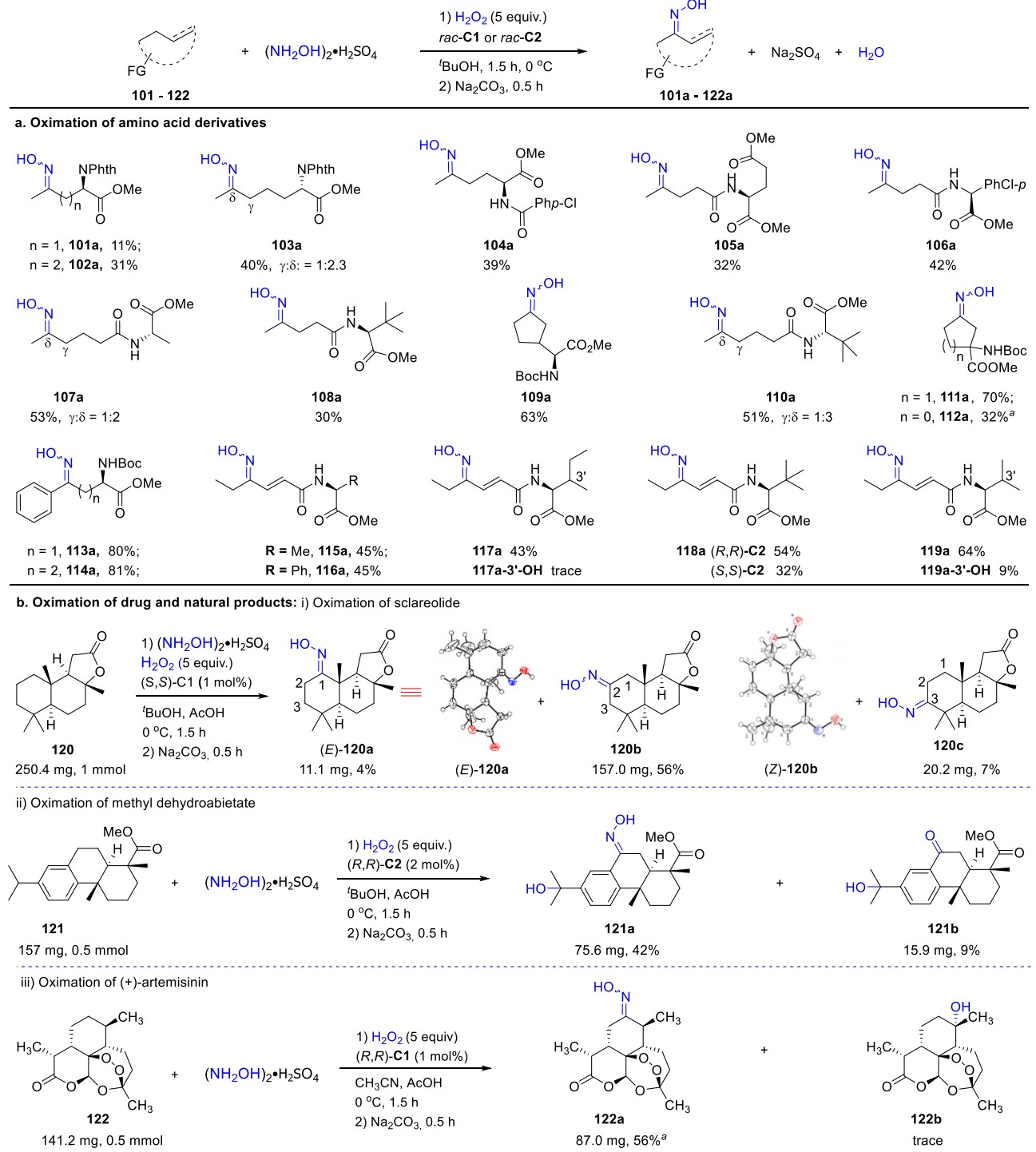

**Fig. 5 | Oximation of multifunctional molecules.** Oximation of amino acid derivatives (**a**) and drug/natural products (**b**, i-iii). General conditions: for substrates **101**–**112**, Fig. 3 conditions used; for **113**–**114**, Fig. 4b conditions used; for **115-119**, Fig. 4a conditions used. Isolated yield reported. [a]Substrate was recycled twice.

(2.0 mL) were added to a reaction tube. The mixture was cooled down to 0 °C in a cryogenic bath, and then $H_2O_2$ (5.0 equiv., 5.0 mmol, 567 µL, 30% wt in $H_2O$) in 2 mL of $^t$BuOH was added with a syringe pump over 1 h under stirring at 0 °C without nitrogen protection. After stirring for an additional 0.5 h, the reaction mixture was quenched with $Na_2SO_3$. Next, the resulting mixture was basified with $Na_2CO_3$ for 0.5 h at 0 °C in a cryogenic bath. After completion of the reaction, water (5.0 mL) was added, and the solution was extracted with dichloromethane (3 × 15.0 mL). The combined organic phase was dried over anhydrous $Na_2SO_4$ and concentrated under reduced pressure to

produce the crude product, which was purified by flash column chromatography to afford the desired product. For further details, please see the Supplementary Information.

## Data availability

All data supporting the findings of this study are available within the paper and the Supplementary Information. Data supporting the findings of this manuscript are also available from the corresponding authors upon request. Crystallographic data are available from the Cambridge Crystallographic Data Centre (CCDC) with the following

codes: ***rac*-C1** (CCDC 2372363); ***meso*-C1** (CCDC 2372362); ***rac*-C2** (CCDC 2372364); ***E*-22a-δ-oxime** (CCDC 2372355); ***E*-65a** (CCDC 2372353); **2*E*,4*E*-119a** (CCDC 2372356); ***E*-120a** (CCDC 2372360); ***Z*-120b** (CCDC 2372361).

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

## Acknowledgements

We gratefully acknowledge the financial support of the State Key Laboratory of Petroleum Molecular & Process Engineering, the Fundamental Research Funds for the National Natural Science Foundation of China (21971156, C.Q.L.; 22578519, X.X.Z), the Shaanxi Provincial Natural Science Foundation (2020JM267, C.Q.L.), and the Fundamental Research Funds for the Central Universities (GK202307002, C.Q.L.). We also thank Professor Zhaotie Liu and group for assistance in propane oxidation, Professor Heyong He and group for discussions, Professor Baoning Zong for advice, and Yan Zhang for assistance in managing peroxides.

## Author contributions

C.Q.L. and J.L.X. conceived and designed the project and wrote the manuscript. H.L. and M.H.S. initiated the study; M.H.S., H.L., X.T.W.L and R.C.W conducted the experiments and analyzed the data. L.X., X.X.Z, A.S., H.M.S. and C.W. contributed to data analysis. All authors have contributed to this work.

## Competing interests

C.Q.L., M.H.S., H.L. and J.L.X. are co-inventors on a Chinese patent (ZL202310609201.0) titled "Diastereoselective green preparation of oxime derivatives by catalytic oxidation of carbon-hydrogen bonds with tetradentate nitrogen-containing coordinated manganese". All other authors declare no competing interests.
