## [Transparent Peer Review file · Nature Communications]

Manganese Catalyzed Oximation of Hydrocarbons to Oximes

Corresponding Author: Professor Jianliang Xiao

Version 0:

Reviewer comments:

Reviewer #1

(Remarks to the Author)

J. Xiao and C. Li, along with their colleagues, have developed a method for the methylene oximation of hydrocarbons using hydrogen peroxide, hydroxylammonium sulfate, and manganese catalysts. This approach enables the synthesis of oximes with good to excellent selectivity and yield across a broad substrate scope. Notably, this catalytic system addresses a long-standing challenge in the field: the direct synthesis of oximes from more abundant and economical hydrocarbons. This advance represents a significant milestone in the field of C-H oxidation catalysis and holds broad interest for researchers in inorganic chemistry, organic chemistry, catalysis, and industry. The manuscript is well-written and clearly presents the research and results. The reviewer recommends the publication of this manuscript in Nature Communications as an article with minor revisions.

1) From an industrial standpoint, the cost-effectiveness of reagents is a critical consideration. Ammonium sulfate is significantly cheaper than hydroxylammonium sulfate, and its use could potentially lower the overall cost of the process. And ammonium sulfate can also be oxidated to hydroxylammonium sulfate under certain oxidation conditions. If ammonium sulfate is used instead of hydroxylammonium sulfate as an amine source, what will be observed in the oxidation of cyclododecane under the reaction conditions?

2) In Figure 2 and Table S-2 (page 24) of the supplementary information, the weakly coordinating anions (OTf⁻, OAc⁻, SO₄²⁻) influence the reactivity and selectivity of the manganese catalyst by affecting its electronic properties and coordination environment. In contrast, non-coordinating anions, such as BF₄⁻ and SbF₆⁻, which do not interact with the catalyst, could alter its behavior by minimizing anion-catalyst interactions. These changes might result in variations in activity, selectivity, or stability. To investigate this further, additional experiments with manganese catalysts utilizing non-coordinating anions (e.g., [Mn(BF₄)₂] or [Mn(SbF₆)₂]) could be conducted. The results from these experiments would be compared with the existing data to elucidate the role of anion coordination in the catalytic cycle. Such insights would enhance our understanding of the reaction's mechanistic aspects and could be incorporated into the revised manuscript.

3) In Figure 2, hydroxylammonium sulfate has a significant effect on the selectivity of oximation of cyclododecane. The role of hydroxylammonium sulfate on affecting could be elucidated. Hydroxylammonium sulfate may influence the regioselectivity for functionalized alkanes by interacting with the functional groups on the substrate or by modulating the reactivity of the manganese catalyst.

To clarify this, the authors could perform additional experiments with functionalized alkanes (e.g., substrates 17 and 30), comparing the regioselectivity of the oximation reaction in the presence and absence of hydroxylammonium sulfate. This would help clarify whether hydroxylammonium sulfate plays a direct role in directing the reaction to specific sites on the substrate. The results could be added in the manuscript or SI.

4) In the SI (Page 13), the numbers next to R groups on the pyridine derivatives should be superscript for consistency and clarity.

Reviewer #2

(Remarks to the Author)

The manuscript presents a one-pot synthesis of oximes from hydrocarbons through manganese-catalyzed oxidation, leading to the formation of ketones, followed by condensation with hydroxylamine. Though similar catalysts have been reported for the oxidation of hydrocarbons, this work shows improved catalytic efficiency for certain substrates (see Figure S-5) by fine-tuning the catalyst structure and optimizing reaction conditions, such as solvent choice and the ratio of H₂O₂ to substrate. The authors also established through control experiments that hydroxylamine sulfate is essential for the selectivity of ketones during the first oxidation step of this reaction. This method can accommodate a range of hydrocarbons, including both activated and inactivated ones, as well as some complex structures. Given the significance of oximes and the direct

functionalization of hydrocarbons, I believe this work merits acceptance in the journal, provided that the following questions are addressed:

1. The statement "the direct synthesis of oximes from hydrocarbons via the oxidative oximation of methylene C-H bonds" is misleading. Initially, I interpreted it as implying that the oximes were formed directly, without the intermediate formation of ketones.
2. The catalyst loadings used are between 1-2 mol%. Is it possible to reduce this, considering the significant improvement in reactivity? One potential application of this reaction is for large-scale production of oximes, and 1 mol% of a metal complex with a sophisticated ligand may not be practical for such applications.
3. Are these catalysts stable throughout the reaction? Additionally, do the silyl group or the Csp³-H bonds in the catalysts participate in the reaction?
4. The role of hydroxylamine sulfate should be clarified, as it is a key difference compared to previous reports on this type of reaction. Could the anion (OTf) surrounding the manganese atom be substituted with (SO₄)²⁻ or (HSO₄)⁻?
5. The E/Z ratios should be included in the main text figures, as they are important and show significantly different reactivities for various configurations. Some ratios in the supporting information are imprecise according to the NMR spectra; for example, the calculation for compound 61a shows 0.91:0.13, which simplifies to 7:1 rather than the reported 9:1. Additionally, how is the E/Z configuration of the oxime assigned? Is the E configuration always the dominant isomer? There may be other factors that affect this ratio, such as hydrogen bonding.
6. Some HRMS data are incorrect: 62a-5'-OH, 115a.
7. Some expressions are incorrect, such as p-CIPh, Php-Br in Fig 3. Ph represents C₆H₅. 99 kcal on page 2 should be 99 kcal.

Reviewer #3

(Remarks to the Author)

The present work by Xiao and co-workers demonstrates the catalytic oxidation of alkanes to carbonyl compounds, followed by oxime formation under controlled conditions using well-defined Mn-PYBP catalytic systems. The catalysts demonstrate a broad substrate scope and high functional group tolerance. The authors claimed that their method offers a superior approach to oxime synthesis and reported an 81% yield for the oxime of cyclohexane. However, they do not address the issue of cyclohexanone formation as a side product. This is analogous to the conventional process, where cyclohexanone must be separated before oximation. If their method still results in a mixture of cyclohexanone and its oxime, then the fundamental inefficiency of requiring separation steps remains unaddressed. A direct comparison of selectivity, separation requirements, and overall process efficiency would be necessary to justify the superiority of their approach. It seems that the primary role of the catalyst is to oxidize the alkanes to the corresponding carbonyl, while basification facilitates the release of free hydroxylamine, which subsequently forms the oxime. The authors have already shown that analogous manganese complexes bearing tetradentate amino ligands can catalyze the benzylic oxidation of alkyl arenes to aryl ketones with H₂O₂ (*Angew. Chem. Int. Ed.* 2022, 61, e202205983). The role of the catalyst in this oximation step is not clear. SI, section 6.3, it is clear that the catalyst is not needed during the oximation step, or if there is any role, it is not mentioned. In fact, the role of the catalyst or the ligand is not clearly mentioned anywhere in the manuscript. While the study presents a potentially valuable oximation method, the lack of mechanistic depth, insufficient control experiments, and novelty make it unsuitable for publication in "Nature Communication" at this stage. The authors should address the following points before submitting this work elsewhere.

1. The influence of hydroxylammonium sulfate on oxidation should be further explored. Specifically, the effect of protonation on hydroxylamine reactivity and oxime coordination to the metal catalyst needs further elaboration.
2. The role of acidic additives in influencing the reaction should be discussed. Why is a higher amount of acid required for challenging acyclic alkanes?
3. The authors stated that "Bypassing the intermediacy of ketones, oximation of the methylene C-H bonds in hydrocarbons via oxidation presents an ideal approach for the construction of oximes," However, their reaction proceeds via the intermediate carbonyl formation. The statement should be checked.
4. The authors stated that "the PYBP ligands with electron-donating, bulky substituents tend to give a higher oxime yield, with the one bearing a 5,5'-triisopropylsilyl group being most effective." Why does a bulky ligand produce a more profound effect?
5. It seems that "rec-C1" exhibits higher compatibility with certain substrates, such as cycloalkanes, while "rac-C2" is more effective for others. Can the authors provide insights into the factors driving this reactivity difference? Specifically, does the ligand environment play a role in influencing substrate selectivity, and if so, how?
6. The manuscript does not provide a catalytic cycle or strong experimental evidence for key intermediates. Radical trapping

experiments (e.g., TEMPO, BHT) should be conducted to confirm whether the reaction follows a radical or non-radical pathway. Kinetic isotope effect (KIE) studies should be performed to understand the rate-determining step. The role of the catalyst and the ligands should be mentioned.

7. SI, Section 6.8, How do the UV-vis measurements support the formation of (rac-L1)Mn(III) and/or (rac-L1)Mn(III)(μ -O)₂Mn(IV) species (65–68) upon mixing rac-1 with H₂O₂, and what role does hydroxylamine sulfate play in accelerating this oxidation process. Can they provide more substantial evidence, such as EPR, to support this claim?

8. What is the oxidation state of the metal in the native complexes. Additional spectroscopic data (such as EPR/XPS) is required to confirm the oxidation state.

9. Why electron-withdrawing substituents deactivate the proximal C-H bond toward oxidation.

10. In SI, Section 6.1, the authors use both "rac-C1" and [Mn], possibly to designate the same catalytic system. It isn't very clear, given that they used other Mn-catalysts. For example, in Table S6, "rac-C2" is the chosen system, yet the corresponding scheme uses "[Mn]". It is difficult to follow which system is being used for a particular work. I would suggest considering replacing [Mn] with "rac-C1", "rac-C2".....

11. In SI, Section 6.2, eq. 1, "NNR" should be replaced with NMR.

Version 1:

Reviewer comments:

Reviewer #1

(Remarks to the Author)

The authors have made necessary improvements to the manuscript. I think the revised version is now acceptable for publication.

Reviewer #2

(Remarks to the Author)

Most of the issues raised in the previous review have been adequately addressed. However, regarding the E/Z isomerism characterization, the authors should clarify whether NOE or other spectroscopic methods were employed to unambiguously assign the configuration. If definitive structural determination is unattainable, this must be explicitly stated in the Supporting Information with a note that only the isomer ratio could be confirmed, not the absolute E/Z configuration. Providing incorrect structural assignments would mislead readers.

Response to Reviewers' comments

We thank all the reviewers for their valuable comments and have revised the manuscript and Supplementary Information (SI) accordingly. In particular, following the comments, more than 100 new experiments have been carried out, which provide more insightful information for the manganese catalyzed oxidative oximation of hydrocarbons to oximes. A major revision arises from the finding that the oximation takes place while the ketone intermediate is being formed instead of post basification as posited in the previous submission (Scheme 1), and the resulting oxime is deactivated towards further C-H oxidation, contributing to enhanced chemoselectivity. Our point-to-point response to the reviewers' comments and explanation concerning Scheme 1 are given below.

Scheme 1. Reaction pathways for the manganese-catalyzed oxidative oximation of hydrocarbons to oximes, proposed based on new experiments conducted during the revision.

Reviewer 1:

a) "From an industrial standpoint, the cost-effectiveness of reagents is a critical consideration. Ammonium sulfate is significantly cheaper than hydroxylammonium sulfate, and its use could potentially lower the overall cost of the process. And ammonium sulfate can also be oxidized to hydroxylammonium sulfate under certain oxidation conditions. If ammonium sulfate is used instead of hydroxylammonium sulfate as an amine source, what will be observed in the oxidation of cyclododecane under the reaction conditions?"

Response:

We thank the reviewer for pointing this out and for the valuable suggestion. Based on the suggestion, we conducted control experiments using ammonium sulfate instead of hydroxylammonium sulfate as the amine source for the oximation of substrate **1** under standard reaction conditions. However, only the ketone products (**1-one** and **1-one'**) were observed, with no formation of the oxime product (**1a**) (see SI, Table S-3, entry 5, page 25). This result indicates that ammonium sulfate cannot be oxidized *in situ* to hydroxylammonium sulfate under our current reaction conditions, highlighting the essential role of hydroxylamine as the direct oxime precursor in the current transformation.

b) "In Figure 2 and Table S-2 (page 24) of the supplementary information, the weakly coordinating anions (OTf⁻, OAc⁻, SO₄²⁻) influence the reactivity and selectivity of the manganese catalyst by affecting its electronic properties and coordination environment. In contrast, non-coordinating anions, such as BF₄⁻ and SbF₆⁻, which do not interact with the catalyst, could alter its behavior by minimizing anion-catalyst interactions. These changes might result in variations in activity, selectivity, or stability. To investigate this further, additional experiments with manganese catalysts utilizing non-coordinating anions (e.g., [Mn(BF₄)₂] or [Mn(SbF₆)₂]) could be conducted. The results from these experiments would be compared with the existing data to elucidate the role of anion coordination in the catalytic cycle. Such insights would enhance our understanding of the reaction's mechanistic aspects and could be incorporated into the revised manuscript."

Response:

We thank the reviewer for the valuable suggestion. Based on the suggestion, we synthesized two additional manganese complexes to investigate the counter anion effect: *rac*-**17** with a coordinating Cl⁻ anion and *rac*-**18** with

a non-coordinating SbF_6^- anion (see SI, section 4.3, pages 20-22). These catalysts were then evaluated alongside *rac*-**1** (with weakly coordinating OTf^- anion) for the oximation of substrate **1** at both 1 mol% and 0.2 mol% catalyst loadings. Notably, *rac*-**18** (SbF_6^-) demonstrated comparable activity to *rac*-**1** (OTf^-), while both showed higher catalytic efficiency than *rac*-**17** (Cl^-) (see SI, Table S-2, entries 32-36, page 24). The observed activity trend (*rac*-**18** \approx *rac*-**1** $>$ *rac*-**17**) demonstrates that non or weakly coordinating anions are necessary in this oximation reaction, most likely for the activation of H_2O_2 at the metal center. Further studies show that not only chloride ion but other potentially coordinating anions, such as phosphate, also inhibit the oxidation reaction (See SI, Table S-3, page 25).

c) “In Figure 2, hydroxylammonium sulfate has a significant effect on the selectivity of oximation of cyclododecane. The role of hydroxylammonium sulfate on affecting could be elucidated. Hydroxylammonium sulfate may influence the regioselectivity for functionalized alkanes by interacting with the functional groups on the substrate or by modulating the reactivity of the manganese catalyst. To clarify this, the authors could perform additional experiments with functionalized alkanes (e.g., substrates **17** and **30**), comparing the regioselectivity of the oximation reaction in the presence and absence of hydroxylammonium sulfate. This would help clarify whether hydroxylammonium sulfate plays a direct role in directing the reaction to specific sites on the substrate. The results could be added in the manuscript or SI.”

Response:

We thank the reviewer for the valuable suggestion. Hydroxylammonium sulfate not only significantly influences the chemoselectivity of cyclododecane oxidation (Figure 2) but also markedly affects that of allylic substrates (see SI, Section 6.9, pages 55-57) and substrate **120** (see SI, Section 6.10, page 58). Additionally, it alters the kinetic isotope effect in the reaction of cyclohexane *vs* cyclohexane- d_{12} (see SI, Section 6.12, pages 59-66). Hydroxylammonium sulfate may play a dual role in the oximation reaction:

- I) **Interaction with the manganese catalyst** to modulate the reaction pathway. Although the mechanistic details of this effect remain unclear, our UV-vis measurements (see SI, Section 6.13, pages 66-72) show that the oxidation of *rac*-**1** with H_2O_2 to higher oxidation states is accelerated by hydroxylammonium cation, but inhibited by free NH_2OH (for effect on catalysis, see SI, Section 6.7, page 52) and excess Cl^- , PO_4^{3-} and to a smaller degree SO_4^{2-} (for effect on catalysis, see SI, Section 5, Table S-3, page 25). The results indicate possible coordination of the hydroxylammonium ion to manganese during oxidation (see SI, Section 6.14, pages 72-73, for a proposed mechanism, and also see response to Reviewer 3).
- II) **Acting as an amine source** to react with ketones, forming oximes and oxime salts that influence reaction selectivity. New experiments were conducted, which revealed that once formed, the oxime becomes much less reactive towards further oxidation compared with the corresponding ketone substrate (see SI, Section 6.4, pages 49-50). The oxime moiety is thus deactivating, contributing to a higher chemoselectivity in oxidative oximation than in ketonization.

Following the reviewer recommendation, we also conducted additional experiments to investigate the regioselectivity of the manganese-catalyzed C–H oxidation of substrates **17** and **30** under conditions without hydroxylammonium sulfate. Comparing the results obtained with *vs* without hydroxylammonium sulfate reveals that the oxime products (**17a** and **30a**) exhibited slightly higher regioselectivities than the corresponding ketone products (**17-one** and **30-one**). These results, which have been included in the updated SI (see SI, Section 6.8, pages 52-55), demonstrate that hydroxylammonium sulfate also plays a role in affecting the regioselectivity in this manganese-catalyzed C–H oxidation system.

d) “In the SI (Page 13), the numbers next to R groups on the pyridine derivatives should be superscript for consistency and clarity”

Response:

We thank the reviewer for pointing this out and have corrected the structure of compounds **L**³⁻¹¹ in Section 4.2 B.

Reviewer 2:

A) “The statement “the direct synthesis of oximes from hydrocarbons via the oxidative oximation of methylene C-H bonds” is misleading. Initially, I interpreted it as implying that the oximes were formed directly, without the intermediate formation of ketones.”

Response:

We thank the reviewer for pointing this out and have revised the main text accordingly.

B) “The catalyst loadings used are between 1-2 mol%. Is it possible to reduce this, considering the significant improvement in reactivity? One potential application of this reaction is for large-scale production of oximes, and 1 mol% of a metal complex with a sophisticated ligand may not be practical for such applications.”

Response:

We thank the reviewer for pointing this out and the valuable suggestion. Based on the suggestion, we further examined the oximation of substrate **1** by reducing the catalyst loading to 0.2 mol% and 0.1 mol% (see SI, Table S-4, pages 25-26). Key findings are:

I) With 0.2 mol% catalyst loading (2 mmol reaction scale, with 4.8 mL of AcOH), we achieved 90% NMR yield and 84% isolated yield of oxime **1a**;

II) At 0.1 mol% catalyst loading (2 mmol reaction scale, with 6.0 mL of AcOH), a 72% NMR yield of **1a** was obtained.

These results demonstrate that the catalyst loading can be successfully reduced to 0.1-0.2 mol%, and suggest that further optimization may enable even lower catalyst loadings.

C) “Are these catalysts stable throughout the reaction? Additionally, do the silyl group or the Csp³-H bonds in the catalysts participate in the reaction?”

Response:

We thank the reviewer for raising this important point and for the insightful question. Both our experimental observations and literature reports (*ACS Catal.* **2023**, *13*, 6403-6414) suggest that the manganese catalysts can undergo deactivation. This might happen through catalyst dimerization, ligand dissociation, and/or oxidation of either the silyl groups or Csp³-H bonds in the ligand framework. However, under our experimental conditions, we did not detect any identifiable degradation products of the catalyst. Being bulky, the silyl group may affect how a substrate, particularly the bulky and/or conformationally flexional one, approaches the Mn=O unit where C-H oxidation occurs, resulting in effects on selectivity (*ACS Catal.* **2020**, *10*, 8611-8631).

D) “The role of hydroxylamine sulfate should be clarified, as it is a key difference compared to previous reports on this type of reaction. Could the anion (OTf) surrounding the manganese atom be substituted with (SO₄)²⁻ or (HSO₄)⁻?”

Response:

We thank the reviewer for pointing this out and the suggestion. Based on our control experiments, hydroxylammonium sulfate not only significantly influences the chemoselectivity of cyclododecane oxidation (Figure 2) but also markedly affects that of allylic methylene oxidation (see SI, Section 6.9, pages 55-57) and substrate **120** (see SI, Section 6.10, page 58). Additionally, it alters the kinetic isotope effect in the reaction of cyclohexane vs cyclohexane-*d*₁₂ (see SI, Section 6.12, page 59-66).

Hydroxylammonium sulfate may play a dual role in the oximation reaction:

- I) **Interaction with the manganese catalyst** to modulate the reaction pathway. Although the mechanistic details of this effect remain unclear, our UV-vis measurements (see SI, Section 6.13, pages 66-72) show that the oxidation of *rac-1* with H₂O₂ to higher oxidation states is accelerated by hydroxylammonium cation, but inhibited by free NH₂OH (for effect on catalysis, see SI, Section 6.7, page 52) and excess Cl⁻, PO₄³⁻ and to a smaller degree SO₄²⁻ (for effect on catalysis, see SI, Section 5, Table S-3, page 25). The results indicate possible coordination of the hydroxylammonium ion to manganese during oxidation (see SI, Section 6.14, pages 72-73, for a proposed mechanism, and also see response to Reviewer 3).
- II) **Acting as an amine source** to react with ketones, forming oximes and oxime salts that influence reaction selectivity. New experiments were conducted, which revealed that the oxime is quickly formed in the oxidation without basification (for details, see our response to Reviewer 3, question A), and once formed, it becomes much less reactive towards further oxidation compared with the corresponding ketone substrate (see SI, Section 6.4, pages 49-50). The oxime moiety is thus deactivating, contributing to a higher chemoselectivity in oxidative oximation than in ketonization.

Based on reviewer questions, we synthesized three manganese catalysts with different anion coordination strengths [*rac-17* (with strong coordinating Cl⁻ anion), *rac-1* (with weak coordinating OTf⁻ anion), and *rac-18* (with non-coordinating SbF₆⁻ anion)] for the oximation of substrate **1**. Notably, *rac-1* (OTf⁻) and *rac-18* (SbF₆⁻) showed comparable catalytic activity, while *rac-17* (Cl⁻) exhibited lower efficiency (see SI, Table S-2, entries 32-36, page 24). These results suggest that under the reaction conditions, the original OTf⁻ anions surrounding the manganese center are most likely replaced by species such as H₂O₂, HOAc and/or hydroxylammonium ion. Note that like Cl⁻ and PO₄³⁻, the anions SO₄²⁻ and HSO₄⁻ inhibit the oxidation when present in excess, presumably *via* weak coordination (see SI, Table S-3, page 25). That this effect is less pronounced in a standard reaction is largely due to the low solubility of the sulfate salts in the reaction system.

Further studies into oxime formation under the reaction conditions are described below in the response to Reviewer 3.

- E) “The E/Z ratios should be included in the main text figures, as they are important and show significantly different reactivities for various configurations. Some ratios in the supporting information are imprecise according to the NMR spectra; for example, the calculation for compound **61a** shows 0.91:0.13, which simplifies to 7:1 rather than the reported 9:1. Additionally, how is the E/Z configuration of the oxime assigned? Is the E configuration always the dominant isomer? There may be other factors that affect this ratio, such as hydrogen bonding.”

Response:

We thank the reviewer for pointing these out and the question. Due to space constraints in the main text including figures and our primary focus being on regioselectivity, we omitted the E/Z ratios in the main text. However, detailed information on both regioselectivity and E/Z ratios for all products can be found in **Section 12** of the SI (pages 102-186). The E/Z ratio indicated has been corrected. Regarding the E/Z configuration assignment, we have included a detailed explanation of the assignment method in the SI (see Section 3, page 8; and Section 12, pages 102-186). For most of the oxime products, E configuration is presumed to be the dominant isomer although caution has been raised (SI, page 8). As the reviewer pointed out, hydrogen bonding may influence the E/Z isomeric ratio of the oxime products. For example, product **100a**, where the oxime hydroxyl group forms hydrogen bonds with the β-amine, is obtained as a single isomer. In contrast, product **113a**, which exhibits no or weaker hydrogen bonding between the oxime hydroxyl group and the γ-amine, exists as a mixture of E and Z isomers (E/Z = 11:1). This difference likely stems from the influence of hydrogen bonding.

F) “Some HRMS data are incorrect: 62a-5'-OH, 115a.”

Response:

We thank the reviewer for pointing this out and have corrected the HRMS data for **62a-5'-OH** and **115a**.

G) “Some expressions are incorrect, such as *p*-ClPh, *Ph**p*-Br in Fig 3. Ph represents C₆H₅. 99 kcal on page 2 should be 99 kcal.”

Response:

We thank the reviewer for pointing this out and have revised the main text.

Reviewer 3:

A) “The authors claimed that their method offers a superior approach to oxime synthesis and reported an 81% yield for the oxime of cyclohexane. However, they do not address the issue of cyclohexanone formation as a side product. This is analogous to the conventional process, where cyclohexanone must be separated before oximation. If their method still results in a mixture of cyclohexanone and its oxime, then the fundamental inefficiency of requiring separation steps remains unaddressed.”

Response:

We thank the reviewer for the comments. There appears to be some misunderstanding, which might arise from our previous submission. Cyclohexanone is not a side product; it is the key intermediate from which oxime results. Our method differs from and advances the conventional process in the sense that oxime is produced from cyclohexane in a one-pot fashion, requiring no separation of the intermediate ketone! The method does not “results in a mixture of cyclohexanone and its oxime”; rather it affords pure oxime upon in-situ basification.

In our initial submission, we asserted, based on NMR analysis, that the formation of the cyclododecanone oxime **1a** “occurs only after basification of the solution”. This assertion has now been proven wrong in the light of a series of new experiments. Our new investigation has shown that the oximation takes place while the ketone intermediate is being formed instead of post basification as posited in the previous submission. This has significant implications, as the resulting oxime is deactivated towards further C-H oxidation, thus contributing to enhanced chemoselectivity. Below is a summary of the results from the new experiments.

- I) A combined quantitative GC and NMR analysis showed that prior to basification, cyclododecanone reacts with hydroxylammonium sulfate, with ca 60% conversion to the oxime **1a** under conditions like those employed for the oxidative oximation of cyclododecane **1** (see SI, Section 6.2, pages 37-40). Through the study, we realized that NMR cannot be reliably used to analyze the oxime salt due to decomposition resulting from sample preparation, nor can GC, due to the incompatible, high boiling point of the sample. To facilitate GC and NMR analysis, we turned attention to cyclohexane and derivatives, which are of lower boiling points and easier to analyze.
- II) By examining the oximation of cyclohexanone with hydroxylammonium sulfate without a base, we have shown that (1) oxime is almost instantly formed, (2) the oximation reaction is catalyzed by Lewis acid, such as Sc(OTf)₃ and presumably by high valent manganese species generated from the oxidation of the catalyst with H₂O₂, and (3) the oxime salt is not detectable on GC (SI, Section 6.3, Table S-17, page 41).
- III) By examining the hydrolysis of cyclohexanone oxime **3a**, we can conclude that (1) oxime undergoes fast hydrolysis to ketone, (2) an equilibrium of the hydrolysis is established almost immediately after subjecting the oxime **3a** to the reaction conditions, and (3) the hydrolysis is facilitated by both Bronsted and Lewis acid, with the later presumably generated by oxidation of the manganese to a higher oxidation state with H₂O₂ (SI, Section 6.3, Tables S-18 and S-19, pages 43-46).
- IV) By following the time course of the oxidative oximation of cyclohexane with GC, we can conclude that the oxime is formed quickly in the oxidative oximation of cyclohexane and formed in a significant amount within half an hour in the oxidation (prior to basification) (SI, Section 6.3, Table S-20, pages 47-48).

B) “A direct comparison of selectivity, separation requirements, and overall process efficiency would be necessary to justify the superiority of their approach. It seems that the primary role of the catalyst is to oxidize the alkanes to the corresponding carbonyl, while basification facilitates the release of free hydroxylamine, which subsequently forms the oxime. The authors have already shown that analogous manganese complexes bearing tetradentate amino ligands can catalyze the benzylic oxidation of alkyl arenes to aryl ketones with H₂O₂ (Angew. Chem. Int. Ed. 2022, 61, e202205983).”

Response:

We thank the reviewer for the comments. As mentioned above, the oxime is formed prior to basification and isolated in a one-pot fashion, involving no separation of the intermediate ketone, and notably, the oxime is much less reactive towards further oxidation compared with the corresponding ketone substrate (see SI, Section 6.4, pages 49-50). The oxime moiety is thus deactivating, contributing to a higher chemoselectivity in oxidative oximation than in ketonization. While the primary role of the catalyst is indeed to enable the oxidation of the alkanes to ketones, it also promotes the oximation of the ketone intermediate (*vide supra*). Our previous work (Angew. Chem. Int. Ed. 2022, 61, e202205983) is about the oxidation of benzylic C–H bonds to aryl ketones, whereas this work provides a general method to access oximes directly from abundant and cost-effective hydrocarbon feedstocks.

C) “The role of the catalyst in this oximation step is not clear. SI, section 6.3, it is clear that the catalyst is not needed during the oximation step, or if there is any role, it is not mentioned. In fact, the role of the catalyst or the ligand is not clearly mentioned anywhere in the manuscript. While the study presents a potentially valuable oximation method, the lack of mechanistic depth, insufficient control experiments, and novelty make it unsuitable for publication in "Nature Communication" at this stage”

Response:

We thank the reviewer for the comments. As mentioned above, the primary role of the catalyst is to enable the oxidation of the alkanes to ketones. In addition, it also promotes the oximation of the ketone intermediate (*vide supra*). We have revised the main text accordingly. We acknowledge, however, that there is a lack of in-depth mechanistic investigation in the study. On the other hand, it is noted that (1) this study is primarily about a new synthetic methodology, (2) the mechanism of the oxidation of hydrocarbons to carbonyls by manganese complexes stabilized by analogous ligands under similar conditions has been widely documented (ACS Catal. 2018, 8, 4528-4538; Inorg. Chem. 2019, 58, 14842-14852; ACS Catal. 2023, 13, 6403-6414; Coord. Chem. Rev. 2019, 384, 126-139; Angew. Chem. Int. Ed. 2020, 59, 7332-7349), (3) we have conducted new experiments (*vide supra*), including experiments suggested by the Reviewer (*vide infra*), which shed light on the reaction pathway including some mechanistic aspects, (4) UV-vis measurements were taken (see SI, Section 6.13, pages 66-72), indicating that the oxidation of *rac-1* with H₂O₂ to higher oxidation states is accelerated by hydroxylammonium cation, but inhibited by free NH₂OH (for effect on catalysis, see SI, Section 6.7, page 52) and excess Cl⁻, PO₄³⁻ and to a smaller degree SO₄²⁻ (for effect on catalysis, see SI, Section 5, Table S-3, page 25). The results indicate possible coordination of the hydroxylammonium ion to manganese during oxidation, and (5) we have suggested a simplified mechanism for the current oxidative oximation in the SI (SI, Section 6.14, pages 72-73).

D) “The influence of hydroxylammonium sulfate on oxidation should be further explored. Specifically, the effect of protonation on hydroxylamine reactivity and oxime coordination to the metal catalyst needs further elaboration.”

Response:

We thank the reviewer for the suggestion. Based on our control experiments, hydroxylammonium sulfate not only significantly influences the chemoselectivity of cyclododecane oxidation (Figure 2) but also markedly affects that of allylic methylene oxidation (see SI, Section 6.9, Page 55-57) and substrate **120** (see SI, Section 6.10, Page 58).

Additionally, it alters the kinetic isotope effect in the reaction of cyclohexane vs cyclohexane-*d*₁₂ (see SI, Section 6.12, Page 59-66).

Hydroxylammonium sulfate may play a dual role in the oximation reaction:

- I) Interaction with the manganese catalyst** to modulate the reaction pathway. Although the mechanistic details of this effect remain unclear, our UV-vis measurements (see SI, Section 6.13, pages 66-72) show that the oxidation of *rac*-**1** with H₂O₂ to higher oxidation states is accelerated by hydroxylammonium cation, but inhibited by free NH₂OH (for effect on catalysis, see SI, Section 6.7, page 52) and excess Cl⁻, PO₄³⁻ and to a smaller degree SO₄²⁻ (for effect on catalysis, see SI, Section 5, Table S-3, page 25). The results indicate possible coordination of the hydroxylammonium ion to manganese during oxidation (see SI, Section 6.14, pages 72-73, for a proposed mechanism, and also see response to Reviewer 3).
- II) Acting as an amine source** to react with ketones, forming oximes and oxime salts that influence reaction selectivity. New experiments were conducted, which revealed that the oxime is quickly formed in the oxidation without basification (*vide infra*), and once formed, it becomes much less reactive towards further oxidation compared with the corresponding ketone substrate (SI, Section 6.4, pages 49-50). The oxime moiety is thus deactivating, contributing to a higher chemoselectivity in oxidative oximation than in ketonization.

We have previously shown that free hydroxylamine (p*K*_a ~ 5.9), i.e. without H₂SO₄, did not afford any oxime in the oxidative oximation of cyclododecane (SI, Section 6.7, page 52), indicating the need for hydroxylamine protonation. The sulfuric acid protonates not only hydroxylamine but also the oxime product (protonated form p*K*_a <1; *Angew. Chem. Int. Ed.* **2008**, 47, 7523–7526) in the solvent used. As mentioned above, free hydroxylamine inhibits the oxidation of manganese catalyst with H₂O₂, presumably via coordination.

- E) “The role of acidic additives in influencing the reaction should be discussed. Why is a higher amount of acid required for challenging acyclic alkanes?”

Response:

We appreciate the reviewer’s insightful question regarding the role of carboxylic acids. Indeed, carboxylic acids are commonly employed as additives to enhance oxidation reaction rates (*ACS Catal.* **2020**, 10, 8611-8631). The acids may function by assisting heterolytic scission of MO-OH bond to form high valent M=O species and stabilize it. In manganese-catalyzed oxidations, a higher acid loading has been shown to improve activity by stabilizing high-valent manganese intermediates, thereby facilitating the oxidation process (*Adv. Synth. Catal.* **2009**, 351, 348-352). For more challenging acyclic alkanes, C–H bond oxidation is expected to be facilitated by high-valent metal–oxo species with sufficient reactivity and lifetime. And presumably, maintaining a higher steady-state concentration of such species can enhance catalytic efficiency. Furthermore, H₂O₂ is more oxidizing under acidic conditions (*E*^o: 1.77 V vs SHE) than basic conditions (*E*^o: 0.87 V vs SHE). In addition, H₂O₂ is less prone to decompose to O₂ under acidic conditions.

- F) “The authors stated that "Bypassing the intermediacy of ketones, oximation of the methylene C-H bonds in hydrocarbons via oxidation presents an ideal approach for the construction of oximes," However, their reaction proceeds via the intermediate carbonyl formation. The statement should be checked.”

Response:

We thank the reviewer for pointing this out and have revised the main text.

- G) “The authors stated that "the PYBP ligands with electron-donating, bulky substituents tend to give a higher oxime yield, with the one bearing a 5,5’-triisopropylsilyl group being most effective." Why does a bulky ligand produce a more profound effect?”

Response:

We thank the reviewer for pointing this out and the question. The manganese catalyst incorporating 5,5'-triisopropylsilyl groups exhibits enhanced steric bulkiness and structural rigidity, which is expected to promote steric isolation of the metal center. This spatial protection could hinder or block catalyst degradation pathways involving, e.g. dimerization. (*ACS Catal.* **2020**, *10*, 8611–8631).

H) “It seems that "rac-C1" exhibits higher compatibility with certain substrates, such as cycloalkanes, while "rac-C2" is more effective for others. Can the authors provide insights into the factors driving this reactivity difference? Specifically, does the ligand environment play a role in influencing substrate selectivity, and if so, how?”

Response:

We thank the reviewer for raising this important point and for the questions. As correctly noted by the reviewer, *rac-C1* demonstrates higher compatibility for oxidizing non-activated C–H bonds. However, for the more active benzylic and allylic substrates, the catalyst led to overoxidation reactions, lowering chemoselectivity to desired oximes.

I) “The manuscript does not provide a catalytic cycle or strong experimental evidence for key intermediates. Radical trapping experiments (e.g., TEMPO, BHT) should be conducted to confirm whether the reaction follows a radical or non-radical pathway. Kinetic isotope effect (KIE) studies should be performed to understand the rate-determining step. The role of the catalyst and the ligands should be mentioned.”

Response:

We thank the reviewer for the valuable suggestions. As recommended, we conducted a radical trapping experiment using TEMPO with substrate **1**, and detected the TEMPO-trapped intermediate (see SI, Section 6.11, pages 58-59). This reaction thus likely involves radical species, which could diffuse to the solution before rebound occurs. Additionally, we performed both competitive and parallel experiments aimed to unearth possible kinetic isotope effect (KIE) using cyclohexane and deuterated cyclohexane-*d*₁₂, both in the presence and absence of hydroxylammonium sulfate. The estimated KIE values are: with hydroxylammonium sulfate: competitive KIE = 2.2, parallel KIE = 2.3; without hydroxylammonium sulfate: competitive KIE = 2.7, parallel KIE = 2.7. These moderate KIE values indicate that C-H bond cleavage may be involved in the rate limiting step of the catalysis, and the hydroxylammonium ion might participate in this step or a step prior to it. The results have been included in the updated SI (SI, Section 6.11, pages 59-66).

Regarding the catalytic cycle and role of catalyst, please see the response above.

J) “SI, Section 6.8, How do the UV-vis measurements support the formation of (rac-L1)Mn(III) and/or (rac-L1)Mn(III)(μ-O)₂Mn(IV) species (65–68) upon mixing rac-1 with H₂O₂, and what role does hydroxylamine sulfate play in accelerating this oxidation process. Can they provide more substantial evidence, such as EPR, to support this claim?”

Response:

We thank the reviewer for pointing this out and the valuable suggestion. Our UV-Vis measurements show that in the presence of hydroxylammonium cation, the absorption of the manganese species in the region of >300 nm increases significantly upon addition of H₂O₂ to a solution of *rac-1* (See SI, Section 6.13, Figs 34-36, pages 67-68). By comparing with the literature data and conclusion [e.g. Fig 9 in *Inorg. Chem.* **2005**, *44*, 3669-3683 and Fig 7 in *Inorg. Chim. Acta.* **2023**, *546*, 121288], our results indicate that the oxidation of *rac-1* with H₂O₂ to higher oxidation states, e.g. (*rac-L*¹)Mn(III) and/or (*rac-L*¹)Mn(III)(μ-O)₂Mn(IV)(*rac-L*¹), is accelerated by hydroxylammonium ion, but inhibited by NH₂OH, Cl⁻, SO₄²⁻ and PO₄³⁻ (See SI, Section 6.13, pages 66-72). As mentioned above, hydroxylammonium cation could coordinate to manganese, presumably via its hydroxy group, which could then

affect the activation of H₂O₂ and the reactivity of the resulting oxo species (see the proposed mechanism for a suggested possibility, SI, Section 6.14, pages 72-73). However, we cannot be sure which oxidation state(s) manganese is oxidized to, and unfortunately, we cannot provide EPR evidence either due to lack of expertise.

K) “What is the oxidation state of the metal in the native complexes. Additional spectroscopic data (such as EPR/XPS) is required to confirm the oxidation state.”

Response:

We thank the reviewer for the question. The molecular structure of *rac*-C1 was determined by X-ray diffraction and is reported in the SI (Section 14, pages 187-194), which is fully consistent with a Mn(II). Following the reviewer’s request, we performed X-ray photoelectron spectroscopy (XPS) analysis, which again confirms the manganese to be in the +2 oxidation state (see SI, Section 13, pages 187).

L) “Why electron-withdrawing substituents deactivate the proximal C-H bond toward oxidation.”

Response:

We thank the reviewer for the question. This effect arises because EWGs reduce the electron density of the C-H bond, thereby hindering hydrogen abstraction by electrophilic metal-oxo species (*Nat. Synth.* **2022**, 1, 682; *Chem. Soc. Rev.* **2022**, 51, 2171.).

M) “In SI, Section 6.1, the authors use both "rac-C1" and [Mn], possibly to designate the same catalytic system. It isn't very clear, given that they used other Mn-catalysts. For example, in Table S6, "rac-C2" is the chosen system, yet the corresponding scheme uses "[Mn]". It is difficult to follow which system is being used for a particular work. I would suggest considering replacing [Mn] with "rac-C1", "rac-C2"...”

Response:

We thank the reviewer for pointing this out and the valuable suggestion, and we have revised the SI.

N) “In SI, Section 6.2, eq. 1, "NNR" should be replaced with NMR”

Response:

We thank the reviewer for pointing this out and have revised the SI.

We thank all the reviewers for their supporting/constructive comments and have addressed the comment of Reviewer 2 (incorporated the comment in the SI, page 8: “Regardless, we can only confirm the isomer ratio, not the absolute E/Z configuration, for these products”).

We have also addressed all the editorial requests.